# Diet-Induced Obesity and NASH Impair Disease Recovery in SARS-CoV-2-Infected Golden Hamsters

**DOI:** 10.3390/v14092067

**Published:** 2022-09-17

**Authors:** François Briand, Valentin Sencio, Cyril Robil, Séverine Heumel, Lucie Deruyter, Arnaud Machelart, Johanna Barthelemy, Gemma Bogard, Eik Hoffmann, Fabrice Infanti, Oliver Domenig, Audrey Chabrat, Virgile Richard, Vincent Prévot, Ruben Nogueiras, Isabelle Wolowczuk, Florence Pinet, Thierry Sulpice, François Trottein

**Affiliations:** 1Physiogenex SAS, F-31750 Escalquens, France; 2Univ. Lille, CNRS, INSERM, CHU Lille, Institut Pasteur de Lille, U1019-UMR 9017-CIIL-Center for Infection and Immunity of Lille, F-59000 Lille, France; 3PLETHA, Institut Pasteur de Lille, F-59000 Lille, France; 4Attoquant Diagnostics, A-1010 Vienna, Austria; 5Sciempath Labo, F-37270 Larçay, France; 6Univ. Lille, INSERM, CHU Lille, Laboratory of Development and Plasticity of the Neuroendocrine Brain, Lille Neuroscience & Cognition, UMR-S 1172, European Genomic Institute for Diabetes (EGID), F-59000 Lille, France; 7Center for Research in Molecular Medicine and Chronic Diseases (CiMUS), S-15781 Santiago de Compostela, Spain; 8Univ. Lille, INSERM, CHU Lille, Institut Pasteur de Lille, U1167-RID-AGE-Facteurs de Risque et Déterminants Moléculaires des Maladies Liées au Vieillissement, F-59000 Lille, France

**Keywords:** obesity, non-alcoholic steatohepatitis, coronavirus disease 2019, SARS-CoV-2, hamster

## Abstract

Obese patients with non-alcoholic steatohepatitis (NASH) are prone to severe forms of COVID-19. There is an urgent need for new treatments that lower the severity of COVID-19 in this vulnerable population. To better replicate the human context, we set up a diet-induced model of obesity associated with dyslipidemia and NASH in the golden hamster (known to be a relevant preclinical model of COVID-19). A 20-week, free-choice diet induces obesity, dyslipidemia, and NASH (liver inflammation and fibrosis) in golden hamsters. Obese NASH hamsters have higher blood and pulmonary levels of inflammatory cytokines. In the early stages of a SARS-CoV-2 infection, the lung viral load and inflammation levels were similar in lean hamsters and obese NASH hamsters. However, obese NASH hamsters showed worse recovery (i.e., less resolution of lung inflammation 10 days post-infection (dpi) and lower body weight recovery on dpi 25). Obese NASH hamsters also exhibited higher levels of pulmonary fibrosis on dpi 25. Unlike lean animals, obese NASH hamsters infected with SARS-CoV-2 presented long-lasting dyslipidemia and systemic inflammation. Relative to lean controls, obese NASH hamsters had lower serum levels of angiotensin-converting enzyme 2 activity and higher serum levels of angiotensin II—a component known to favor inflammation and fibrosis. Even though the SARS-CoV-2 infection resulted in early weight loss and incomplete body weight recovery, obese NASH hamsters showed sustained liver steatosis, inflammation, hepatocyte ballooning, and marked liver fibrosis on dpi 25. We conclude that diet-induced obesity and NASH impair disease recovery in SARS-CoV-2-infected hamsters. This model might be of value for characterizing the pathophysiologic mechanisms of COVID-19 and evaluating the efficacy of treatments for the severe forms of COVID-19 observed in obese patients with NASH.

## 1. Introduction

Coronavirus disease 2019 (COVID-19, caused by severe acute respiratory syndrome coronavirus 2 (SARS-CoV-2)) is still a global health problem, with millions of infected individuals and deaths worldwide [1]. In most individuals, SARS-CoV-2 infections are generally asymptomatic or induce only mild symptoms. However, the infection can lead to severe pneumonia, acute respiratory distress syndrome, and death in patients with risk factors such as old age, male sex, hypertension, diabetes, chronic lung disease, heart, liver, and kidney diseases, and cancer [2]. Importantly, non-alcoholic fatty liver disease (NAFLD) and obesity are known to be major risk factors for severe COVID-19 [3,4].

The links between obesity, NAFLD, and severe COVID-19 can be viewed in the context of greater urbanization, low physical activity, and constantly available, energy-dense food, which are driving a dramatic increase in the prevalence of obesity and type-2 diabetes worldwide [5]. The prevalence of NAFLD is also increasing because the disease is strongly correlated with both obesity and type-2 diabetes [6]. NAFLD covers a spectrum of disorders, ranging from liver steatosis to non-alcoholic steatohepatitis (NASH). NASH is associated with inflammatory infiltrates, hepatocyte “ballooning”, and fibrosis, all of which increase the likelihood of disease progression to cirrhosis and hepatocellular carcinoma [7]. Although the causal nature of the relationships has not yet been confirmed, NAFLD and liver fibrosis are also associated with an elevated risk of cardiovascular events and functional cardiac abnormalities [8].

It is still not clear how the abovementioned metabolic comorbidities drive the severity of COVID-19. Adipose tissue inflammation and immune impairments might contribute to disease severity [9] and compromise vaccine efficacy in obese individuals [10]. Furthermore, angiotensin II (the pro-inflammatory and pro-fibrotic effector of the activated renin-angiotensin system (RAS)) might influence the severity of COVID-19 [11]. Today’s vaccines are available to prevent SARS-CoV-2 infection and/or to lower the severity of COVID-19, and several drugs with antiviral and/or anti-inflammatory activities are also being repositioned or developed for the prevention or treatment of severe COVID-19 [12]. Preclinical animal models are required to rapidly validate these therapeutic strategies and thus counter the still-present threat of COVID-19. However, the use of non-human primates as preclinical animal models raises ethical concerns and has a low throughput [13]. Since the SARS-CoV-2 spike protein does not bind to the murine angiotensin-converting enzyme 2 (ACE2) entry receptor, the transgenic expression of human ACE2, or virus adaptation, is required in murine models. This can result in major differences; for example, SARS-CoV-2 infection leads to lethal encephalitis in transgenic human ACE2-expressing mice, which limits the translation of the results to the clinic [14]. The impact of obesity on COVID-19 outcomes has been studied in C57BL/KsJ-*db*/*db* mice, using a mouse-adapted SARS-CoV-2 strain [15]. However, leptin receptor deficiency does not replicate the human context, i.e., the prevalence of leptin receptor deficiency is only 0–3% [16]. Furthermore, mice do not have the same lipid/lipoprotein metabolism [17,18], bile acid profile [19], and diet-induced NASH characteristics as humans; for example, hepatocyte ballooning is not observed in these animals [20]. These differences further limit the use of murine models for the evaluation of the putative links between COVID-19, obesity, and NASH. In contrast, the golden hamster has emerged as a useful model of SARS-CoV-2 infection.

The SARS-CoV-2 spike protein binds to hamster ACE2, and the resulting lung disease shares several features with COVID-19 [21,22,23]. In the golden Syrian hamster model, however, SARS-CoV-2 infection does not lead to severe clinical symptoms [22,24]. This experimental model can be improved through the use of nutritional approaches. For instance, relative to controls, the dyslipidemia and fatty liver induced by a Western (i.e., high-fat, high-sugar) diet increase the severity of the lung disease and delay functional lung recovery from a SARS-CoV-2 infection [25]. The latter study did not describe the impact of SARS-CoV-2 infection in terms of liver damage. To better replicate the human context, a better preclinical hamster model (i.e.**,** developing obesity, NASH, and liver fibrosis) is urgently needed. We have recently established the free-choice-diet-induced obese hamster model, which develops obesity, dyslipidemia, NASH, and liver fibrosis [26,27]. Our recent data validated the interest in developing this preclinical model of COVID-19 [28]. Relative to the control animals, albeit with an identical viral load, more lung and liver inflammation developed during the acute phase of SARS-CoV-2 infection in this hamster model of obesity with metabolic comorbidities. In the present study, we investigated the long-term effects of SARS-CoV-2 infection on obese NASH hamsters. The results indicate that obesity and NASH impair disease recovery in SARS-CoV-2-infected hamsters (higher levels of lung fibrosis and failure to recover the initial body weight) and sustain the dyslipidemic, inflammatory profile and liver damage, which are indicative of long-term sequelae. Hence, this nutritional model might be valuable for the further investigation of mechanisms leading to sustained disease outcomes and drug efficacy in order to control COVID-19 in obese NASH patients.

## 2. Materials and Methods

### 2.1. Animals, Diet, Virus, and Infection

Male golden Syrian hamsters (age at the start of the study: 4 weeks) were purchased from Janvier Labs (Le Genest-St-Isle, France). A first set of experiments were performed on non-infected animals (Experiment #1, Appendix A). After a five-day acclimation period, ten golden hamsters were fed for up to 20 weeks with either a control chow diet (5.1% fat, 19.3% protein, 55.5% carbohydrates, minerals 4.6%, fiber 4%, and humidity 11.5%, SAFE Diets, Augy, France) with access to regular laboratory animal drinking water, or a free-choice diet (Appendix A). The latter consisted of a choice (within the same cage) between control chow and regular drinking water or a previously described high-fat/high-cholesterol diet with vegetable oils as the fat source (55% control chow (SAFE Diets), 20% peanut butter paste (Skippy, Hormel Foods Corporation, Austin, MN, USA) and 25% hazelnut paste (Nustikao, Leclerc, Ivry-sur-Seine, France); 40.8% fat, 14.8% protein, 44.4% carbohydrates, and 0.5% cholesterol, overall) and 10% fructose-enriched drinking water [26]. After 20 weeks of the diet, hamsters were fasted for 6 h and anesthetized with isoflurane. A retro-orbital blood sample was collected for biochemical assays. The animals were then euthanized and exsanguinated with saline prior to organ collection (the lung and the liver) for biochemical assays and histologic assessments.

In a second set of experiments (Experiment #2, Appendix A), thirty golden hamsters were fed with either a control chow diet or a free-choice diet for 20 weeks and then transferred to the Institut Pasteur de Lille biosafety level 3 (BSL3) facility (Lille, France) for infection with SARS-CoV-2. Animals were infected intranasally with a sublethal dose of the SARS-CoV-2 clinical isolate BetaCoV/France/IDF/0372/2020 strain (Wuhan origin), supplied by the French National Reference Center for Respiratory Viruses (hosted by the Institut Pasteur, Paris, France) [29]. The SARS-CoV-2 isolate had been passaged three times on Vero cells and once on Vero E6 cells before the infection. The virus had been checked by sequencing to ensure that the original seed virus had not mutated. Animals were anesthetized by the intraperitoneal injection of 300 µL of phosphate-buffered saline (PBS), containing ketamine (100 mg/kg), atropine (0.75 mg/kg), and Valium (2.5 mg/kg), and were then infected intranasally with 100 µL of Dulbecco’s Modified Eagle’s medium (DMEM) with or without SARS-CoV-2 (2 × 10^4^ 50% tissue culture infectious dose, TCID50). The TCID50 corresponded to the viral titer at which 50% of the infected cells displayed a cytopathic effect. Body weight was monitored after SARS-CoV-2 infection. All animals were infected and kept in isolators within the BSL3 facility. For the tissue collection, animals were euthanized by the intraperitoneal injection of 140 mg/kg pentobarbital sodium. Blood, lung, and liver samples were collected from non-infected (mock) hamsters and from SARS-CoV-2-infected hamsters 4, 7, 10, and 25 days post-infection (dpi). The right lobes of the lung were used to quantify the viral load and gene expression levels and the left lobe was used in histologic analyses. Between 5 and 8 lean and obese NASH hamsters were analyzed per time point.

### 2.2. Determination of the Viral Load and the Median Tissue Culture Infectious Dose

Infectious viral titers and viral RNA levels were, respectively, quantified with the Reed & Muench TCID50 assay and quantitative reverse-transcription PCR (RT-qPCR) tests. The titration of live infectious virus was performed as follows. Tissue was homogenized in Lysing Matrix D tubes containing 1 mL of PBS, using the mixer mill MM 400 (Retsch, Haan, Germany) (15 min, 15 Hz). After centrifugation at 15,000× *g* for 5 min, the clarified supernatant was harvested for live virus titration. For the TCID50 assay, the supernatant was serially diluted in DMEM containing 1% penicillin/streptomycin, and the dilutions were transferred onto Vero E6 cells on 96-well plates. Briefly, serial 10-fold dilutions of each sample were inoculated onto a Vero E6 cell monolayer in duplicate and cultured in DMEM supplemented with 2% fetal bovine serum (Invitrogen, Waltham, MA, USA) and 1% penicillin/streptomycin and L-glutamine. The cultures were observed for cytopathic effects daily for 5 or 6 days. Viral RNA in the lung tissue was quantified as follows. Briefly, the tissue was homogenized in 1 mL of RA1 buffer from the NucleoSpin RNA Kit, containing 20 mM of tris(2-carboxyethyl) phosphine (Macherey Nagel, Hoerdt, France). Total RNA in the tissue homogenate was extracted with the NucleoSpin RNA Kit (Macherey Nagel, Hoerdt, France) and eluted in 60 µL of water.

### 2.3. Assessment of the Gene Expression (RT-qPCR Assays)

RT-qPCR assays was performed on lung and liver samples collected on dpi 0, 4, 7, and 10. The RNA was reverse-transcribed with the High-Capacity cDNA Archive Kit (Life Technologies, Courtaboeuf, France). The resulting cDNA was amplified using SYBR Green-based real-time PCR and QuantStudio™ 12K Flex Real-Time PCR systems (Applied Biosystems™, Villebon-sur-Yvette, France), according to the manufacturer’s instructions. The expression was normalized against the *ACTG1* housekeeping gene encoding γ-actin. Specific primers were designed using Primer Express software (Applied Biosystems) and ordered from Eurofins Scientifics (Ebersberg, Germany) (Table 1). Relative mRNA levels (2^−∆∆Ct^) were determined by comparing the respective PCR cycle thresholds (Ct) for the gene of interest and the housekeeping gene (∆Ct) and those for the infected vs. mock groups (∆∆Ct). Viral RNA loads were normalized against γ-actin expression levels (∆Ct).

### 2.4. Histopathology

Lung tissue was fixed in PBS with 4% formaldehyde for 7 days, rinsed in PBS, transferred to a 70% ethanol solution, and embedded into paraffin blocks. Tissue sections (thickness: 4 µm) were stained with hematoxylin and eosin (H&E) and scanned with a Nanozoomer (Hamamatsu Photonics, Massy, France). Morphologic changes in the lungs were rated semi-quantitatively using a dual histopathology scoring system adapted from Imai et al. [30] and Meyerholz and Beck [31]. A total of nine parameters ((i) cellular death/necrosis, (ii) alveolar and/or perivascular edema, (iii) hyaline membranes or fibrin, (iv) inflammation, (v) thrombi, (vi) congestion, (vii) hemorrhage, (viii) type-II hyperplasia, and (ix) syncytia) were qualitatively assessed with a score from 0 to 4 (0 = absent; 1 = 1–10% of the lung section affected; 2 = 11–25% affected; 3 = 26–50% affected; 4= >50% of the lung section affected) and ranked. To evaluate the presence of pulmonary fibrosis on dpi 0 and 25, the Sirius-Red-stained area on scanned sections from Experiment #2 was measured using a computer-assisted, automated, whole-section histomorphometric image analysis technique (Visiopharm, Hørsholm, Denmark). The morphometric evaluation was performed on virtual whole sections at a magnification of 20 (corresponding to 0.46 μm/pixel). An algorithm for Sirius Red morphometric measurements of lung-stained sections was generated with the Bayesian linear segmentation tool in the software package, which was further refined through training on a subset of sections from 5 animals. Major histology section artefacts, such as large vascular and peribronchiolar structures, and alveolar lumen were automatically dissected and removed from the area of interest. The Sirius-Red-positive area (in mm^2^) was measured and expressed as a percentage of the area of interest. The accuracy of the morphometric evaluation was checked for each individual image. The liver histology assessments (H&E and Sirius Red staining), NAFLD activity scoring, and Sirius Red labeling have been described previously [27].

### 2.5. Biochemical Assays and Measurement of Circulating ACE2, ACE, and Renin Activities

Serum samples were assayed with a Pentra 400 machine and the corresponding assay kits (Horiba France SAS, Longjumeau, France), as described previously [26,27]. Serum equilibrium levels of angiotensin peptides (Ang I, Ang II, Ang-[1–7], and Ang-[1–5]) were measured using a liquid chromatography–tandem mass spectrometry (LC–MS/MS) technique (Attoquant Diagnostics, Vienna, Austria), as described previously [32]. Serum ACE activity (Ang II/Ang I) and serum renin activity (Ang I + Ang II) were calculated from the angiotensin peptide levels. Serum ACE2 activity was measured using a fluorometric assay kit (BioVision, Milpitas, CA, USA). Commercial ELISA kits were used to assay interleukin (IL)-6 (Fine Test, Wuhan, China) and monocyte chemoattractant protein-1 (MCP-1) (MyBiosource, San Diego, CA, USA).

### 2.6. Statistical Analyses

The data are expressed as the mean ± standard deviation (SD) unless stated otherwise. Statistical analyses were performed using GraphPad Prism software (version 6). Data were tested for normality using the D’Agostino–Pearson normality test. An unpaired, two-tailed Student’s *t*-test or a Mann–Whitney U test was used to compare the group of lean hamsters with the groups of obese NASH hamsters. Three or more groups were compared in a one-way analysis of variance, followed by Dunn’s post-test or the nonparametric Kruskal–Wallis test. Sidak’s multiple comparison test was used to detect significant differences in the percentage change from the initial body weight. The threshold for statistical significance was set to *p* < 0.05.

## 3. Results

### 3.1. Free-Choice-Diet-Induced Obesity in Hamsters Is Associated with Dyslipidemia, NASH, and Inflammation

Compared with the chow-fed animals, the golden hamsters fed on a free-choice diet for 20 weeks developed metabolic disorders, which were characterized by excess body weight (mean value: 42% greater than in lean hamsters) and marked dyslipidemia, with higher serum levels of total cholesterol, high-density lipoprotein (HDL) cholesterol, low-density lipoprotein (LDL) cholesterol, and triglycerides (Table 2). The free-choice-diet-fed hamsters also presented elevated serum alanine aminotransferase (ALT) levels, indicative of liver damage and/or inflammation. Compared with the lean animals, the obese hamsters had heavier livers and higher total cholesterol, triglyceride, and fatty acid levels, features that are indicative of liver steatosis (Table 2). Indeed, a histologic assessment of the liver (Figure 1A–D) and the histopathology score (Table 2) confirmed the presence of hepatic steatosis and inflammation in the obese hamsters but not in the lean hamsters. Obese hamsters also displayed hepatocyte ballooning, as evidenced by cytokeratin-18 immunostaining (Figure 1E,F). Relative to the lean hamsters, obese hamsters had a substantially higher fibrosis score (ranging from a score of 2 for perisinusoidal/periportal fibrosis to a score of 3 for bridging fibrosis), as reflected by the higher percentage of Sirius Red labeling (Table 2 and Figure 1C,D). Overall, the total NAFLD activity score was markedly higher in obese hamsters than in lean hamsters, confirming that free-choice-diet-induced obesity was associated with NASH and liver fibrosis. Free-choice-diet-induced obesity and NASH were also associated with elevated systemic inflammation, as reflected by higher serum MCP-1 and IL-6 levels (Figure 1G). Interestingly, the obese NASH hamsters also presented elevated protein levels of IL-6 in the lungs (Figure 1H). Taken as a whole, these data show that a 20-week free-choice diet induces obesity, dyslipidemia, liver inflammation, and fibrosis (i.e.**,** NASH) in the golden hamster. The blood and lung levels of inflammatory cytokines were higher in the obese NASH hamsters than in lean hamsters. Collectively, the obese NASH hamsters displayed not only the expected metabolic comorbidities (dyslipidemia and NASH/liver fibrosis) but also a pro-inflammatory profile with elevated serum levels of IL-6 and MCP-1 in the steady state, both of these inflammatory biomarkers being linked to the severity of COVID-19 in humans.

### 3.2. Obese NASH Hamsters Show High Levels of Lung Damage and Fibrosis and Fail to Recover Their Initial Body Weight during the Late Stages of SARS-CoV-2 Infection

Our recent data indicate that obese NASH hamsters develop more lung inflammation during the acute phase response of SARS-CoV-2 infection (until dpi 10) [28]. Here, we sought to determine the long-term effects of a SARS-CoV-2 infection on free-choice-diet-induced obese NASH hamsters. Lean and obese NASH animals showed substantial body weight loss, starting on dpi 1 and peaking on dpi 7 (Figure 2A). The two groups did not differ with regard to weight loss and clinical symptoms (including ruffled fur and labored breathing). Whereas the lean hamsters had returned to their initial body weight by dpi 25, obese NASH hamsters had not, suggesting a weaker recovery from SARS-CoV-2 infection. On dpi 4, the lung viral load (according to a TCID50 assay) was similar in the two groups of animals (Figure 2B), in line with our previous findings [28]. No infectious virus particles (in a TCID_50_ assay) were detected on dpi 7 and thereafter (data not shown). These data were confirmed by quantification of the viral RNA-dependent RNA polymerase (RdRp) expression in a qRT-PCR assay on dpi 4 (Figure 2C). Trace amounts of viral RNA were detected on dpi 7 (Figure 2C) but not on dpi 10 or 25 (data not shown). In line with these findings, the lean hamsters and obese NASH hamsters did not differ in their expression of the interferon (IFN)-stimulated genes (ISGs) *Isg15*, the IFN-induced GTP-binding protein *Mx1* on dpi 4 and 7 (Figure 2D and Appendix A), and the anti-inflammatory genes *Infg*, *Il10*, and *Il6* (Appendix A).

We next examined the signs of lung disease in SARS-CoV-2-infected animals. By dpi 4, both groups had developed acute bronchointerstitial pneumonia, with moderate necrotizing bronchiolitis, congestion, discrete alveolar hemorrhage, intra-alveolar and interstitial infiltrations by neutrophils and macrophages, and bronchiolar epithelial cell death/necrosis (Appendix A). On dpi 7, more than 75% of the lung sections showed signs of subacute bronchointerstitial pneumonia. This was characterized by extensive and intense hyperplasia of the bronchiolar epithelial cells and type-II alveolar pneumocytes, as well as interstitial, perivascular, and intra-alveolar mixed-cell inflammation. Infection was also associated with alveolar collapse, some hyaline membranes, hemorrhage, microvasculitis, and discrete thrombi. On dpi 4 and 7, the histopathologic score was similar in lean hamsters and obese NASH hamsters (Figure 2I). On dpi 10, the histopathologic damage ranged from subacute to chronic (Figure 2E,F). In lean hamsters, we observed mild to- moderate hyperplasia of the type-II pneumocytes and inflammation due to lymphocytes and macrophages. The pulmonary changes were similar in obese NASH hamsters, although the inflammation, syncytial and alveolar hemorrhages, and hyperplasia of type-II pneumocytes were more extensive and more severe than in lean hamsters. On dpi 10, the histopathologic score was higher in obese NASH hamsters compared to lean hamsters (Figure 2I). On 25 dpi, the histopathologic scoring was markedly lower and had a more chronic profile than at earlier time points for both lean hamsters and obese NASH hamsters (Figure 2I). The lesions corresponded to mild or very mild perivascular, peribronchiolar, and septal lymphohistiocytic inflammation, thickening of the inter-alveolar septa, and mild or very mild hyperplasia of type-II pneumocytes. Lung fibrosis was assessed as the percentage of areas stained by Sirius Red dye (Figure 2G,H). In the steady state, the level of lung fibrosis was slightly higher in obese NASH hamsters than in lean hamsters (Figure 2J and Appendix A). Infection with SARS-CoV-2 enhanced Sirius Red labeling in both the lean and NASH obese groups. Interestingly, on dpi 25, relative to the infected lean hamsters, lung fibrosis was markedly higher in obese NASH hamsters, with Sirius Red staining lining the alveolar ducts and inter-alveolar septa (Figure 2G,H,J).

Taken as a whole, the results show that obese NASH hamsters and lean hamsters displayed similar viral loads and levels of lung inflammation during the early phase of a SARS-CoV-2 infection. At later time points, however, the recovery phase was delayed in the obese NASH hamsters. This delay translated to higher levels of lung fibrosis and failure to recover the initial body weight, which are indicative of long-term sequelae.

### 3.3. The Dyslipidemic, Pro-Inflammatory Profile of Obese NASH Hamsters Is Sustained during SARS-CoV-2 Infection

Compared with the lean hamsters, obese NASH hamsters had higher serum ALT levels before SARS-CoV-2 infection (day 0) (Table 2 and Figure 3A). The serum ALT levels in lean hamsters did not rise over the course of infection, whereas a nonsignificant trend towards elevated serum ALT levels was observed on dpi 4 and 7 in obese NASH hamsters, which was suggestive of greater liver damage during SARS-CoV-2 infection. In humans, obesity and NASH are associated with elevated circulating levels of free fatty acids (FFAs) [33]. Serum FFA levels were significantly higher in the obese NASH hamsters than in lean animals both before SARS-CoV-2 infection and afterwards (except on dpi 25) (Figure 3B). Although the infection had no impact on the FFA levels in lean hamsters, a significant increase was observed on dpi 7 in the obese NASH hamsters. Serum triglyceride levels were higher at baseline in the obese NASH group but had fallen significantly by dpi 4 in both groups (Figure 3C). Total cholesterol levels did not change significantly during the course of the infection but were significantly higher in the obese NASH hamsters than in lean hamsters (data not shown). While the obese NASH hamsters had significantly higher baseline levels of anti-atherogenic HDL cholesterol, SARS-CoV-2 infection was associated with significantly reduced HDL cholesterol concentrations in both the lean hamsters and obese NASH hamsters on dpi 4 and 7 (Figure 3D). Furthermore, SARS-CoV-2 infection was associated with an increase in the pro-atherogenic LDL cholesterol levels in obese NASH hamsters (but not in lean hamsters) on dpi 4 and 7 (Figure 3E). Both the HDL cholesterol and LDL cholesterol levels had returned to basal levels in the lean and obese NASH animals by dpi 25. In line with the systemic inflammation associated with obesity and NASH, the blood levels of the pro-inflammatory cytokine MCP-1 were significantly higher in the obese NASH hamsters than in lean hamsters both before and after SARS-CoV-2 infection. The MCP-1 levels were found to be increased on dpi 4 and 7 in obese NASH animals, but the increase was not statistically significant (Figure 3F). The IL-6 concentration did not change over the course of the infection (not shown).

Taken as a whole, our results indicate that obese NASH hamsters infected with SARS-CoV-2 sustained pro-inflammatory and dyslipidemic profile, with higher FFA and LDL cholesterol levels and lower HDL cholesterol levels.

### 3.4. SARS-CoV-2 Infection in Obese NASH Hamsters Is Associated with Differences in Serum RAS Activity and Notably Higher Levels of Angiotensin II

ACE2 (the receptor for SARS-CoV-2 [34]) is a potent inhibitor of RAS activation, which involves a cascade of vasoactive and inflammatory peptides [11]. ACE2 cleaves the vasoconstrictive, proinflammatory peptide angiotensin II into the vasodilatory, anti-inflammatory peptide angiotensin (1-7) [11]. The effect of SARS-CoV-2 infection on circulating ACE2 activity in humans is subject to debate. The activity was found to stay constant in some studies [32,35] and to increase in others [36]. Compared with the lean hamsters, obese NASH hamsters had a significantly lower baseline level of circulating ACE2 activity (Figure 4A). Whatever the timepoint post-infection, SARS-CoV-2 had no impact on the circulating ACE2 activity, which remained lower in the obese NASH hamsters than in the controls. The ACE activity was then calculated by measuring the equilibrium levels of angiotensin peptides using an LC-MS/MS technique. The circulating ACE activity increased (albeit not significantly) during the early stages of the infection and was slightly but significantly higher (*p* = 0.002) in the obese NASH animals than in lean controls on dpi 25 (Figure 4B). We next quantified the serum renin activity, the rate-limiting step in the conversion of angiotensin I to angiotensin II. At baseline, the serum renin activity was slightly but not significantly higher in the obese NASH hamsters than in lean hamsters (Figure 4C and Table 3). The activity fell as early as dpi 4 in both groups and remained low until dpi 25, at which time it was slightly higher in the obese NASH hamsters than in lean hamsters. The lower level of serum renin activity during SARS-CoV-2 infection explains the substantial fall in the levels of all the circulating angiotensins (Table 3). It is noteworthy that the angiotensin II levels (which fell during infection) were higher in the obese NASH hamsters than in lean animals before and after infection (Figure 4D).

### 3.5. Obese NASH Hamsters Show Inflammation, Fibrosis, and Impaired Liver Metabolism after SARS-CoV-2 Infection

We next explored the potential effects of SARS-CoV-2 infection on the liver metabolism, NASH, and fibrosis. Compared with the lean animals, obese NASH hamsters expressed high mRNA levels of fatty acid synthase (Fasn) in the liver (Figure 5A). Although the hepatic level of Fasn mRNA remained constant during the infection in lean hamsters, it decreased strikingly in the obese NASH hamsters. In contrast, the mRNA expression of cholesterol 7α-hydroxylase (Cyp7a1, the rate-limiting enzyme in bile acid metabolism) had increased significantly in the lean hamsters by dpi 25 (and, to a lesser extent, by dpi 7) but remained constant in obese NASH hamsters (Figure 5B). In the obese NASH hamsters, in line with Sencio et al. [28], SARS-CoV-2 infection was associated with greater hepatic mRNA expression of the pro-inflammatory factors IL-1β, MCP-1, and IL-6 (particularly on dpi 4 and 7). Figure 5C–E shows the fold inductions relative to the mock-infected animals. In contrast, SARS-CoV-2 infection did not modulate the hepatic mRNA expression of these pro-inflammatory factors in the lean animals. On dpi 4, the hepatic expression of the profibrogenic cytokine TGF-β was significantly greater in the obese NASH hamsters than in lean controls (Figure 5F). As expected, H&E and Sirius Red staining revealed that the steady-state liver lesions were more intense and widespread in the obese NASH hamsters, i.e., with steatosis, inflammation, hepatocyte ballooning, and fibrosis (representative histology images are shown in Appendix A, and scores are shown in Figure 5K). In NAFLD patients, weight loss of at least 10% results in the resolution of non-alcoholic steatohepatitis and a reduction in fibrosis by at least one stage [37]. It is noteworthy that, despite the early weight loss induced by SARS-CoV-2 infection and the incomplete body weight recovery, exposure to the virus was not associated with a lower NAFLD activity score in the obese NASH hamsters (Figure 5G,H,K). Furthermore, levels of hepatic fibrosis were substantially higher in the obese hamsters than in lean hamsters (Figure 5I,J,L). Taken as a whole, the data indicate that SARS-CoV-2 infection in the obese NASH hamsters was associated with greater inflammation and fibrosis and differences in the liver metabolism (gene expression).

## 4. Discussion

Recent clinical studies have shown that obesity and NASH influence the severity of COVID-19 [3,4]. The hamster is a valuable preclinical model of SARS-CoV-2 infection and suitably replicates some of the metabolic features seen in humans with underlying conditions such as obesity, dyslipidemia, and NASH. Although the hamster model shares some features with the human model with regard to lung disease [22,23,24], the use of “healthy” hamsters (with normal physiologic parameters) limits our understanding of the potential links between metabolic comorbidities and the severity of COVID-19. We have recently proposed that obese NASH hamsters might be a valuable model of COVID-19, at least during the acute phase response of SARS-CoV-2 infection [28]. The present study describes the main long-term consequences of SARS-CoV-2 infection in obese NASH hamsters.

We sought to determine whether metabolic disorders could predispose hamsters to severe COVID-19 and worse recovery. The diet-induced obese hamster model displayed not only the expected metabolic comorbidities (dyslipidemia and NASH/liver fibrosis) but also a pro-inflammatory profile (with elevated serum levels of IL-6 and MCP-1) in the steady state. Both of these inflammatory biomarkers have been linked to the severity of COVID-19 in humans. It has been suggested that the IL-6 level is an independent predictor of the disease risk [38], while MCP-1 has been linked to a higher mortality rate [39]. We next investigated the consequences of SARS-CoV-2 infection in obese NASH hamsters. The lung viral loads on dpi 4 were similar in the lean hamsters and obese NASH hamsters. Both groups showed a similarly robust antiviral (ISG-related) response, as reported previously [28]. This might explain why the groups’ lung histopathologic scores during the early phase of infection were also similar. However, the resolution of the lung damage was delayed in the obese NASH hamsters on dpi 10 (relative to the controls), and a high level of pulmonary fibrosis was evidenced on dpi 25. These findings are in line with Port et al.’s report, stating that Western-diet-induced dyslipidemic (non-obese) hamsters had higher levels of lung damage and delayed functional lung recovery after a SARS-CoV-2 infection [25]. In the latter model, a delayed viral clearance and, in line with the present manuscript, a delayed body weight recovery was reported. The reason for this delayed recovery in our setting is not clear. One can hypothesize that, along with pre-existing metabolic comorbidities, the lower systemic levels of ACE2 activity and the higher angiotensin II levels drive a shift towards pro-inflammatory and pro-fibrotic pathways [40]. Pulmonary fibrosis is a potential sequela of COVID-19 survivors. This raises the question of whether antifibrotic treatments might mitigate post-viral fibrosis [41]. The potential benefits of these treatments could perhaps be evaluated using the obese NASH hamster model.

In humans, dyslipidemia is associated with more severe COVID-19 and a higher mortality rate, although the underlying mechanisms have yet to be elucidated [42]. In contrast to the mouse [17,18], the golden hamster differs less from humans with regard to the lipid and lipoprotein metabolism [43]. Hence, the hamster model is appropriate for investigating the impact of dyslipidemia on the severity of COVID-19. During SARS-CoV-2 infection, obese NASH hamsters showed a sustained dyslipidemic profile. The elevated serum FFA levels observed in obese NASH hamsters might especially favor liver damage during the infection. In humans, serum FFA levels are correlated with the presence of NAFLD [33]. In diet-induced obese rats, FFAs and/or their metabolites are responsible for liver damage via an elevation in oxidative stress [44]. The higher levels of pro-atherogenic LDL cholesterol seen in obese NASH hamsters might favor the production of oxidized LDL particles, which in turn might stimulate inflammation and apoptosis [45]. In hamsters, inflammation and infection also induce LDL oxidation in vivo [46]. Furthermore, the low levels of HDL cholesterol in the obese NASH hamsters seen on dpi 4 and 7 might limit the HDL cholesterol’s anti-inflammatory, antithrombotic, anti-oxidant, and anti-apoptotic effects [47].

The RAS is believed to play a role in severe COVID-19 [11]. Although the blood equilibrium RAS peptide concentrations in patients with COVID-19 have been reported, the potential impacts of metabolic disorders (dyslipidemia) and/or liver disease have not been investigated [35]. The present study is the first to show that SARS-CoV-2 infection in obese NASH hamsters influences the serum RAS peptide concentrations. In line with clinical studies [36], SARS-CoV-2 infection in the hamsters was associated with a dramatic drop in the serum angiotensin peptide levels, including those of the protective RAS components angiotensin 1-5 and angiotensin 1-7. It is noteworthy that the enzymatic activities of ACE and ACE2 (the two main RAS modulators) were not affected by the infection, whereas the level of renin activity fell dramatically (in line with Kutz et al.’s clinical observations [35]). Interestingly, there were significant differences in the levels of angiotensins I and II (the central players in the RAS) between the obese NASH hamsters and lean hamsters. It remains to be seen whether this difference influences lung and liver damage and, thus, the severity of COVID-19.

The diet-induced preclinical model developed in the present study is of value for evaluating the impacts of COVID-19 on NASH and liver fibrosis. One of the major findings was the sustained high NAFLD activity score of obese NASH hamsters during SARS-CoV-2 infection, despite substantial weight loss (~18% on dpi 7) and incomplete body weight recovery. Hence, the weight loss induced by SARS-CoV-2 infection does not lead to a reduction in liver damage in obese NASH hamsters. In parallel, the expression of pro-inflammatory genes was upregulated in the liver, especially in the early stages of the SARS-CoV-2 infection. Although low ACE2 protein levels in the liver of obese NASH hamsters might well favor a pro-inflammatory and profibrotic profile [48], the underlying mechanisms remain unclear. It remains to be determined how the changes in the expression of the *Fasn* (involved in FFA synthesis) and *Cyp7a1* (involved in bile acid synthesis) genes in obese NASH hamsters influence the liver metabolism, NASH, and fibrosis. At present, there are no conclusive data for humans. The hypothetical mechanisms of liver damage in COVID-19 patients include the direct infection of the hepatocytes by SARS-CoV-2 and indirect effects of systemic inflammation [49,50,51]. In the model developed in the present study, an RT-PCR assay did not detect any viral RNA-dependent RNA polymerase transcripts (data not shown). SARS-CoV-2 infection impairs the insulin/insulin-like growth signaling pathway genes [52]. It would be interesting to determine whether this pathway is more significantly altered in infected obese NASH hamsters relative to infected lean animals.

The present study had a number of limitations. Firstly, the hamster model reproduces some but not all of the features of COVID-19; the hamster is infected by SARS-CoV-2 but survives and rapidly resolves the COVID-19-like disease, although post-acute sequelae persist [53]. Moreover, it does not replicate the non-respiratory symptoms (particularly those affecting the intestines) observed in patients with severe COVID-19. People with type-2 diabetes and COVID-19 require more medical interventions and are significantly more likely to die [54]. As the present diet-induced obese NASH hamster model does not develop type-2 diabetes, per se [26,27], it might be interesting to investigate the impact of SARS-CoV-2 infection on hamsters with streptozotocin-induced diabetes [55]. Lastly, we assessed the serum markers during the infection and can only speculate that the measured circulating concentrations (e.g., those of the RAS components) are associated with tissue concentrations and disease outcomes. Despite these limitations, we reported on a novel obese NASH hamster model, in which metabolic comorbidities, a pro-inflammatory profile, altered ACE2 activity, and higher angiotensin II levels were associated with impaired recovery from a SARS-CoV-2 infection. As the hamster model (lean) has recently been demonstrated to be an attractive model of long COVID-19 [53], this preclinical model of obesity and NASH should be of value for investigating the mechanisms of long COVID-19 in co-morbid individuals and for evaluating novel therapies mitigating long-term sequelae, both in terms of pulmonary and extra-pulmonary dysfunction and metabolism.

## Figures and Tables

**Figure 1 viruses-14-02067-f001:**
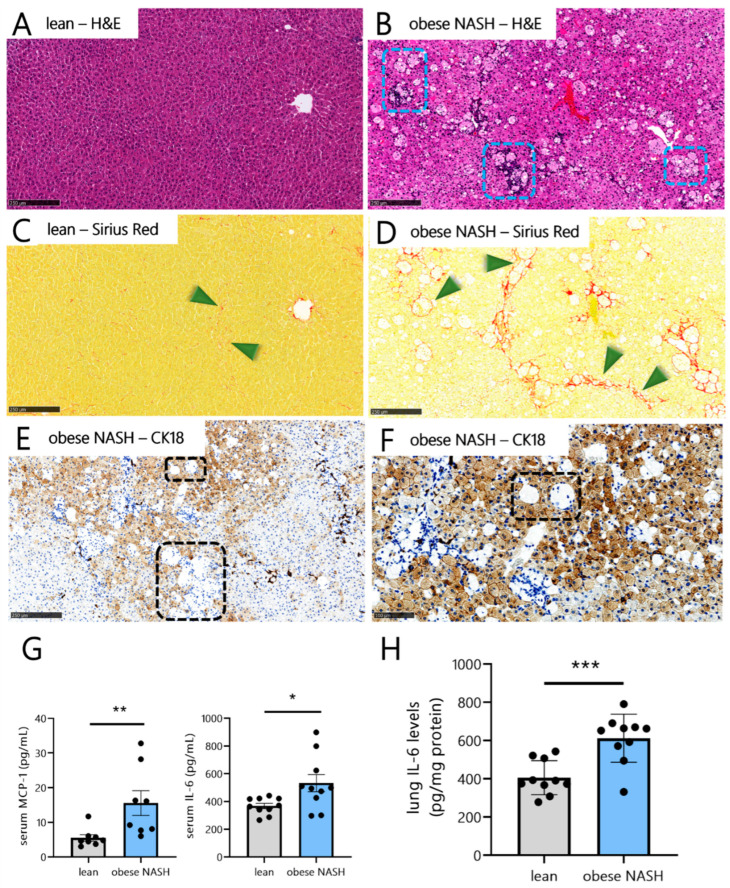
Free-choice-diet-induced obesity, NASH/liver fibrosis, and lung inflammation. (**A**–**D**) Representative liver histology images after H&E staining (**A**,**B**) and Sirius Red staining (**C**,**D**) of lean hamsters (**A**,**C**) and obese NASH hamsters (**B**,**D**), respectively. Blue dashed squares indicate inflammatory foci and hepatocyte ballooning, and green arrows indicate liver fibrosis. (**E**) Cytokeratin-18 (CK18) immunostaining of liver samples from obese NASH hamsters. Black dashed squares indicate clusters of ballooned hepatocytes lacking cytokeratin-18. (**F**) Hepatocyte ballooning is shown at a higher magnification. (**G**) Serum levels of MCP-1 and IL-6 and (**H**) lung levels of IL-6 protein in lean hamsters and obese NASH hamsters. Data are shown as the mean ± SD (*n* = 10 animals per group). Significant differences were determined using an unpaired, two-tailed Student’s *t*-test or a Mann–Whitney U test (* *p* < 0.05; ** *p* < 0.01; *** *p* < 0.001). Scale bars: 250 µm (**A**–**F**).

**Figure 2 viruses-14-02067-f002:**
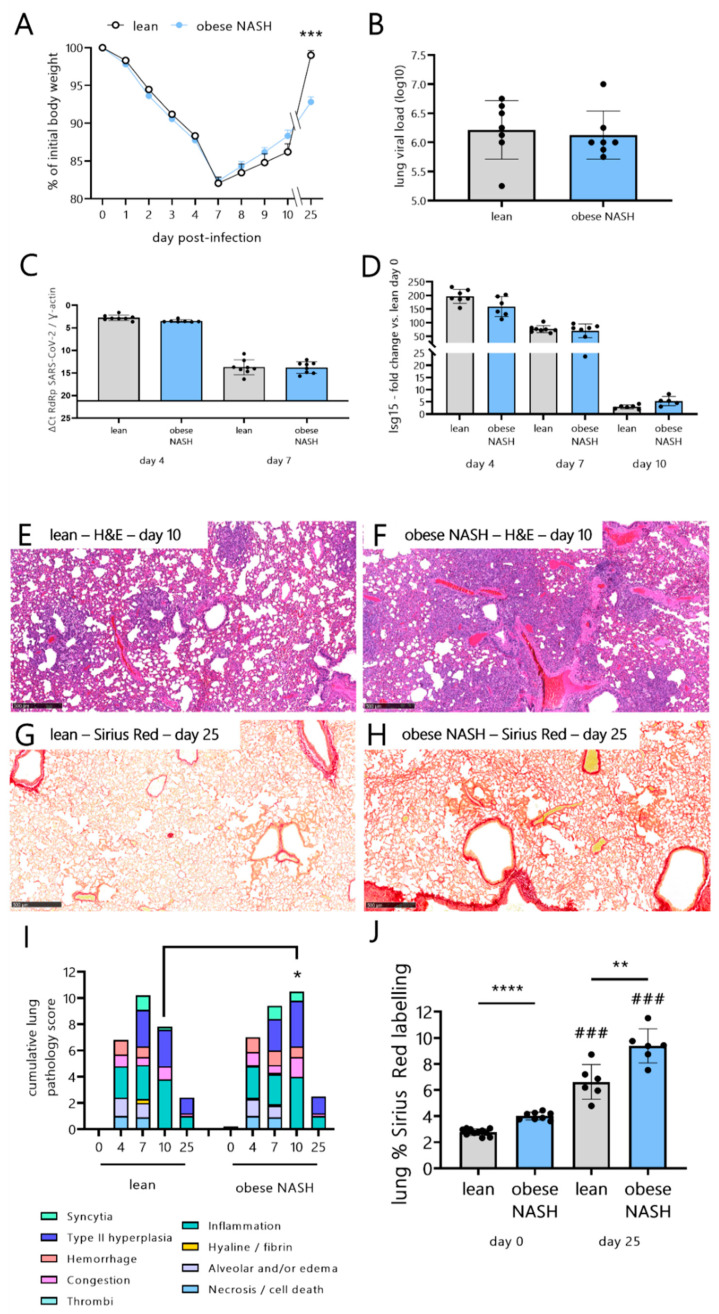
Impact of SARS-CoV-2 infection on lean hamsters vs. obese NASH hamsters. Hamsters were inoculated intranasally with 2 × 104 TCID50 of the clinical SARS-CoV-2 isolate hCoV-19/France/lDF0372/2020. (**A**) Changes in body weight (as the mean ± SD percentage of the initial body weight). Sidak’s correction for multiple comparisons was used to detect significant differences (*** *p* < 0.001). (**B**) Determination of the infectious viral load in the lungs on dpi 4 (TCID50 assay). The data are expressed as the number of infectious virus particles per lung. (**C**) Viral RNA-dependent RNA polymerase (RdRp) transcript levels in the lungs, as quantified in a qRT-PCR (dpi 4 and 7). The data are expressed as the delta Ct (normalized against γ actin). The bottom line corresponds to the detection threshold. The results are expressed as the mean ± SD. (**D**) Isg15 mRNA transcripts were quantified by qRT-PCR (lung). The data are expressed as the mean ± SD fold increase over the level observed in the mock-treated lean hamsters. It should be noted that there were no significant differences between the mock-treated lean hamsters and mock-treated obese NASH hamsters (day 0) (not shown). (**A**–**D**) *n* = 5–8 animals/group. (**E**,**F**) Histopathologic assessment of lung sections from lean hamsters (**E**) and obese NASH hamsters (**F**) on dpi 10. Representative images after H&E staining) are depicted. (**G**,**H**) Representative lung histology images after Sirius Red staining (25 dpi). (**I**) The lung histopathologic score following SARS-CoV-2 infection (on dpi 4, 7, 10, and 25) (*n* = 5–8 animals/group). (**J**) The percentage of lung tissue labeled by Sirius Red on day 0 and on dpi 25. The data are quoted as the mean ± SD or (panel I) the median (*n* = 6–11 animals/group). Significant differences were determined using the Mann–Whitney U test (* *p* < 0.05; ** *p* < 0.01, *** *p* < 0.001, **** *p* <0.0001). The values of the various groups were also compared with those of mock-infected animals (### *p* < 0.001). Scale bars: 500 µm (**E**–**H**).

**Figure 3 viruses-14-02067-f003:**
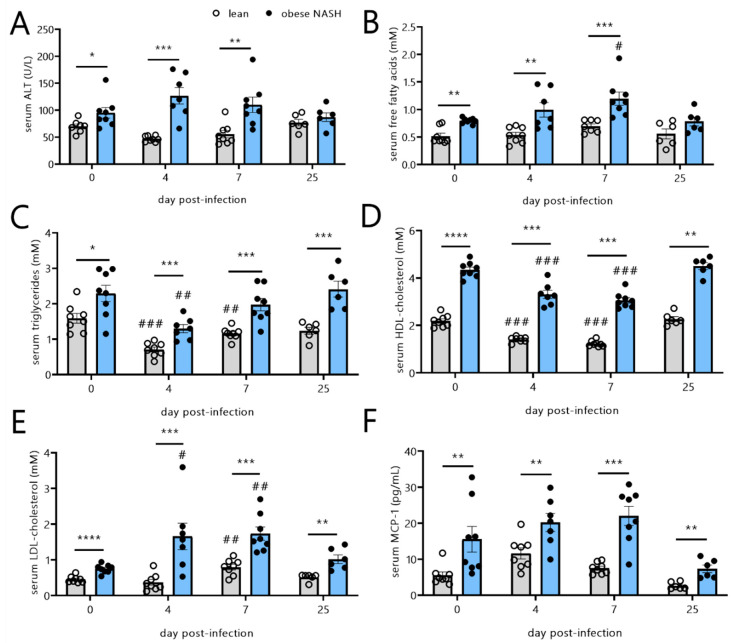
Effects of SARS-CoV-2 infection on serum variables in lean hamsters vs. obese NASH hamsters. Blood from the mock-infected and SARS-CoV-2-infected hamsters was collected at different timepoints post-infection. Serum levels of (**A**) ALT, (**B**) free fatty acids, (**C**) triglycerides, (**D**) HDL cholesterol, (**E**) LDL cholesterol, and (**F**) MCP-1 on dpi 4, 7, and 25 are shown. The data are expressed as the mean ± SD (n = 6–8 animals/group). Significant differences between lean hamsters and obese NASH hamsters were determined using the Mann–Whitney U test (* *p* < 0.05; ** *p* < 0.01, *** *p* < 0.001, **** *p* < 0.0001). The effect of SARS-CoV-2 infection is also depicted. The values were compared with those of mock-infected animals using a Kruskal–Wallis test and Dunn’s post-test (# *p* < 0.05, ## *p* < 0.01, and ### *p* < 0.001).

**Figure 4 viruses-14-02067-f004:**
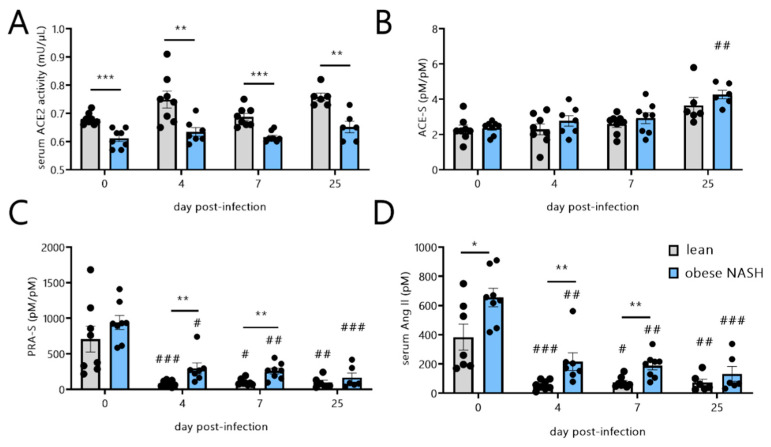
Effects of SARS-CoV-2 infection on serum ACE2, ACE, and the renin-angiotensin system activity levels in lean hamsters vs. obese NASH hamsters. Blood from mock-infected and SARS-CoV-2-infected lean hamsters and obese NASH hamsters were collected at the indicated timepoints post-infection. Serum ACE2 (**A**), ACE (**B**), and renin angiotensin (**C**) activities are depicted. Angiotensin II levels are shown in (**D**). The data are expressed as the mean ± SD (*n* = 6–8 animals/group). Significant differences were determined using the Mann–Whitney U test (* *p* < 0.05; ** *p* < 0.01, *** *p* < 0.001). The effect of SARS-CoV-2 infection is also depicted. The values were compared with those of mock-infected animals using a Kruskal–Wallis test and Dunn’s post-test (#*p* < 0.05, ## *p* < 0.01, and ### *p* < 0.001). ACE-S, angiotensin-converting enzyme activity (Ang II/Ang I); PRA-S, serum renin activity (Ang I + Ang II).

**Figure 5 viruses-14-02067-f005:**
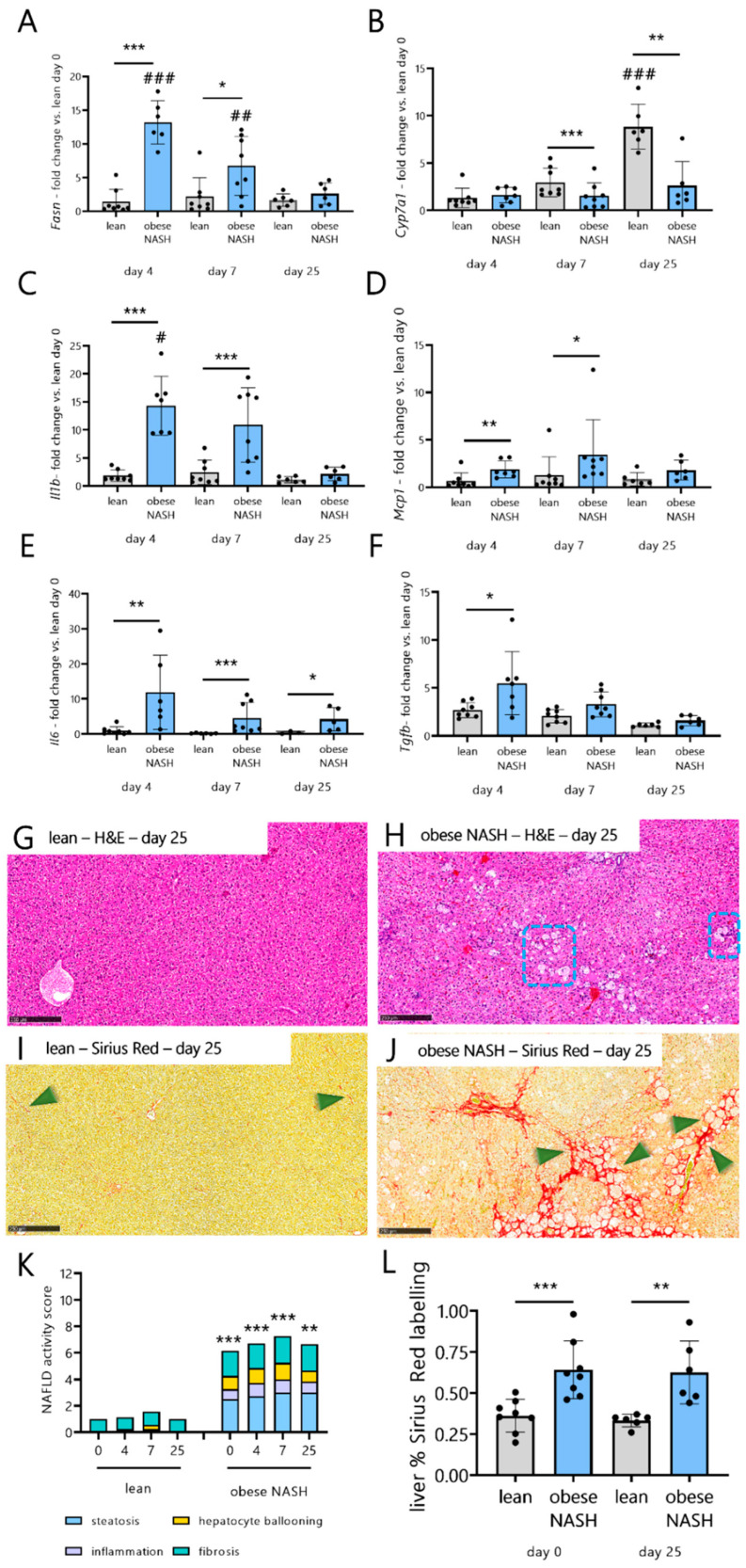
Effects of SARS-CoV-2 infection in terms of liver damage and fibrosis in lean hamsters vs. obese NASH hamsters. The livers of mock-infected and SARS-CoV-2-infected hamsters were collected on dpi 4, 10, and 25. (**A**–**F**) mRNA copy numbers were quantified by qRT-PCR. The data were expressed as the fold increase over the expression level in mock-treated lean animals. (**G**–**J**) Representative liver histology images after H&E staining (**G**,**H**) or after Sirius Red staining (**I**,**J**) of lean hamsters and obese NASH hamsters. (**K**) The total NAFLD activity score. (**L**) The percentage of the liver section stained by Sirius Red on day 0 and dpi 25. Data are shown as the mean ± SD (*n* = 6–8 animals/group). Significant differences were determined using the Mann–Whitney U test (* *p* < 0.05; ** *p* < 0.01, *** *p* < 0.001). The effect of SARS-CoV-2 infection is also depicted. The values of SARS-CoV-2-infected hamsters were compared with those of mock-infected animals using a Kruskal–Wallis test and Dunn’s post-test (# *p* < 0.05, ## *p* < 0.01, and ### *p* < 0.001). Scale bars: 250 µm (**G**–**J**).

**Table 1 viruses-14-02067-t001:** The qPCR primer sequences.

Gene	GenBank ID	Forward Sequence (5′-3′)	Reverse Sequence (5′-3′)
*Ccl2*	XM_005076967.4	TGCTAACTTGACGCAAGCTCC	AAGTTCTTGAGTCTGCGGTGG
*Cyp7a1*	XM_005066730	AGCAACTGACTGTGCCTAGGAAA	GGAACTCAGGCAGTGAGAACAGA
*Fas*	XM_013112078	GGCAACTCCTGGTATGTTCACTTC	CCTTCTGGCCATTTTACCTTTTCT
*Il1b*	XM_005068610	GAAGTCAAAACCAAGGTGGAGTTT	TCTGCTTGAGAGGTGCTGATGT
*Il6*	XM_005087110	CCATGAGGTCTACTCGGCAAA	GACCACAGTGAATGTCCACAGATC
*Il10*	XM_005079860.3	GGTTGCCAAACCTTATCAGAAATG	TTCACCTGTTCCACAGCCTTG
*Infg*	NM_001281631.1	TGTTGCTCTGCCTCACTCAGG	AAGACGAGGTCCCCTCCATTC
*Isg15*	XM_013119951.3	CTG GTG CCC CTG ACT AAC T	CTG TCA TTC CGC ACC AGG AT
*Mx1*	XM_021229467.1	GGTATCGTTACCAGGTGCCC	GGTCTGGAACACTTGGGGAG
*RdRp*	/	GTGARATGGTCATGTGTGGCGG	CARATGTTAAASACACTATTAGCATA
*Tgfb1*	XM_013125593	CGGGATCAGCCTCAAACG	TGAGGAGCAGGAAGGGTCTGT
*Srebf1*	XM_005067680	TTAGGGACCTTTGTCACTGGCT	AGGTCGGCATGATCCTGATT

**Table 2 viruses-14-02067-t002:** Clinical and biochemical parameters in lean and obese NASH hamsters. Data are shown as mean ± SD, *n* = 10 hamsters per group. * *p* < 0.05, ** *p* < 0.01, *** *p* < 0.001, **** *p* < 0.0001 lean vs. obese NASH. Statistical analysis was performed using an unpaired 2-tailed Student *t*-test or a Mann–Whitney test.

Parameter	Lean	Obese NASH
Body weight (g)	137 ± 9	194 ± 11 ****
Serum total cholesterol (mmol/L)	2.68 ± 0.43	5.57 ± 0.79 ****
Serum HDL cholesterol (mmol/L)	2.14 ± 0.47	4.70 ± 0.91 ****
Serum LDL cholesterol (mmol/L)	0.40 ± 0.22	0.79 ± 0.20 ***
Serum triglycerides (mmol/L)	1.61 ± 0.53	3.92 ± 1.22 ****
Serum ALT (U/L)	97± 17	128 ± 35 *
Serum AST (U/L)	81 ± 29	75 ± 38
Liver weight (g)	5.4 ± 0.4	10.7 ± 1.0 ****
Hepatic total cholesterol (mmol/g liver)	14.7 ± 2.1	157.6 ± 20.6 ****
Hepatic triglycerides (mmol/g liver)	12.8 ± 2.7	58.9 ± 12.7 ****
Hepatic fatty acids (µmol/g liver)	6.5 ± 1.3	21.9 ± 4.4 ****
Hepatic Steatosis score	0.0 ± 0.0	2.3 ± 0.5
Inflammation score	0.0 ± 0.0	1.0 ± 0.0
Ballooning score	0.0 ± 0.0	1.0 ± 0.0
Fibrosis score	1.0 ± 0.0	2.6 ± 0.5 ****
Liver %Sirius Red labelling	0.31 ± 0.03	0.52 ± 0.05 **
Total NAFLD score	1.0 ± 0.0	6.9 ± 0.6 ****

**Table 3 viruses-14-02067-t003:** Equilibrium levels of circulating angiotensin peptides. Values are expressed in mean ± SD (*n* = 6–8 animals/group).

Angiotensin Peptides	Angiotensin I (1-10) (pmol/L)		Angiotensin II (1-8) (pmol/L)		Angiotensin 1-7 (pmol/L)		Angiotensin 1-5 (pmol/L)	
Days Post-infection/Diet	Lean	Obese	Lean	Obese	Lean	Obese	Lean	Obese
0	218.5 ± 147.4	285.3 ± 112.3	488.9 ± 369.3	655.0 ± 177.6	77.5 ± 45.0	81.2 ± 27.1	108.6 ± 63.7	99.7 ± 24.7
4	24.2 ± 11.1	79.8 ± 51.2	56.0 ± 32.0	215.5 ± 159.3	12.8 ± 4.9	21.4 ± 12.9	11.8 ± 5.1	21.3 ± 9.0
7	27.8 ± 9.4	68.7 ± 35.6	73.9 ± 36.3	188.1 ± 82.9	9.9 ± 5.7	19.9 ± 10.9	13.7 ± 6.1	21.5 ± 12.0
25	23.4 ± 22.8	33.5 ± 32.8	71.9 ± 58.0	131.8 ± 123.6	11.0 ± 9.8	12.3 ± 11.0	17.4 ± 11.4	31.9 ± 28.9

## Data Availability

The data generated in this study were deposited at https://doi.org/10.5281/zenodo.

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
