# Peer review of "Diet-Induced Obesity and NASH Impair Disease Recovery in SARS-CoV-2-Infected Golden Hamsters"

_viruses, 2022, doi:10.3390/v14092067_

Round 1
Reviewer 1 Report
The study aims to explore the associations between obesity and COVID-19 sequela and severity. The authors used golden hamsters as experimental model which is appropriate for preclinical study because spike protein capable of binding to ACE2 of hamster. The post-viral response of liver, lung , inflammatory markers and RAS system were well described in the main text. In general, the study was well designed and getting novelty, the section of outcome and discussion were informative, and the graphs was well illustrated . However, I still have some minor concerns.
1. Why the all hamsters survive the COVID , even with prolonged sequela left
2. Kidney injuries during COVID were reported in the literature in human and animal models. NASH was also associated with CKD in human. Did the authors ever assess renal function and the non-recovery of body weight and high Ang-II of obese NASH hamsters may be due to azotemia or CKD-related anorexia ?
3. insulin/IGF pathway impairment in COVID also plays crucial role in the obese liver pathogenicity.
Author Response
We would like to thank the reviewers for their thorough evaluation and fair comments. Our point-by-point responses to the comments received are summarized below, with the reviewer's original question italicized, and our response in bold.
Reviewer #1
The study aims to explore the associations between obesity and COVID-19 sequela and severity. The authors used golden hamsters as experimental model which is appropriate for preclinical study because spike protein capable of binding to ACE2 of hamster. The post-viral response of liver, lung , inflammatory markers and RAS system were well described in the main text. In general, the study was well designed and getting novelty, the section of outcome and discussion were informative, and the graphs was well illustrated . However, I still have some minor concerns.
Thank you very much for this comment.
- Why the all hamsters survive the COVID , even with prolonged sequela left
In this study, we used a sublethal dose of virus. As pointed out by the reviewer, obese NASH animals survive infection. As shown in Figure 2, differences develop during the chronic phase of the disease and not during the acute phase, the period during which animals might die from infection.
- Kidney injuries during COVID were reported in the literature in human and animal models. NASH was also associated with CKD (Xic kidney dis )in human. Did the authors ever assess renal function and the non-recovery of body weight and high Ang-II of obese NASH hamsters may be due to azotemia or CKD-related anorexia ?
This is a very interesting point but unfortunately, the NSB3 environment did not allow us to measure these criteria, i.e. chronic kidney diseases, change in proteinuria and CKD-related anorexia, as we cannot use metabolic cages to collect urine.
- insulin/IGF pathway impairment in COVID also plays crucial role in the obese liver pathogenicity.
This is also an interesting remark but to study potential alteration of the the insulin/IGF pathway in this model, it would require a study in its own (e.g. RNAseq etc). We have now indicated in the revised form (page 17) that it may be interesting to investigate the insulin/IGF pathway in this model.

Reviewer 2 Report
Review of the manuscript „Diet-induced obesity and NASH impair disease recovery in SARS-CoV-2-infected golden hamsters“
I have read with great interest the manuscript in which the authors described the results of the SARS-CoV-2 infection influence on outcomes of metabolic, inflammatory, and liver parameters of diet-induced obese golden Syrian hamsters regarding stages of SARS-CoV-2 infection and compared to the control group of golden hamsters. The study found that obese hamsters had worse recovery than lean counterparts, and had sustained liver steatosis, inflammation, hepatocyte ballooning, and noticeable liver fibrosis. The authors concluded that in hamsters, diet-induced obesity and NASH impaired SARS-CoV-2 disease recovery. The animal model of golden hamsters was used since the SARS-CoV-2 spike protein binds to hamster ACE2, and the resulting lung disease shares several features with COVID-19 which is comparable to humans. However, in the golden Syrian hamster model, SARS-CoV-2 infection does not lead to severe clinical symptoms. Still, the authors believe that this model might be of value in characterizing the pathophysiologic mechanisms of COVID-19 and in evaluating the efficacy of treatments for the severe forms of COVID-19 observed in obese patients with NASH. Overall, the manuscript is written good, with good data and future direction.
However, I have the following suggestions for manuscript improvement:
I suggest writing the manuscript in the passive form, and to exclude the “we” or “our” form in sentences. I believe that the paper could sound more scientifically if it is all written in the passive.
Methods
The Methods section seems vaguely presented. I suggest that the section be broken down into "Participants" (Animals) and Methods with details of the diet used and methods used for the analysis of outcome parameters. In that section, it is certainly important to include the number of tested hamsters, its total number, the number of hamsters in the intervention group and the number of hamsters in the control group. In the manuscript, there is no listed number of tested hamsters. I suppose that all hamsters had access to (and free to use) a control diet and to intervention diet (used for inducing obesity). It is not clearly stated how many hamsters were obese after 20 weeks (diet-induced obese hamsters) and how many hamsters were used for the control group. It can be suggested to display a diagram of the research flow so that the research flow can be seen.
When describing a control diet (chow) used for feeding hamsters used by SAFE Diets, Augy, France, it is stated that it consisted of 1% fat, 19.3% protein, and 55.5% carbohydrates. What were other ingredients up to 100%? I suggest stating other ingredients as "other non-energy ingredients" or something similar or recalculating the proportions of macronutrients with regard to energy content (100%). Described intervention diet has listed proportions of ingredients up to 100%, I suggest describing the control diet similarly.
The phrase “dpi” (days post-infection) in Methods should be written in full meaning since there is the first time mentioned.
In Statistical Analysis, I suggest mentioning that data were previously tested for normality using the appropriate test since the authors mention that some data were parametric and some non-parametric.
Results
The authors presented very well their study results mostly in the form of Figures, which I very much respect. If study results were presented more in tables, with means (SD), medians (IQR) and p-values, future research can value that data presented this way. I appreciate the intention of the authors to present their data in the form of Figures, they give a picture of the rise or fall of certain values, regardless of their significant difference, but it is difficult to read the values. For example, the authors mentioned that HDL-cholesterol levels dropped on days 4 and 7 post-infection, it is difficult to see was the drop higher on day 4 or on day 7. Also, there is a rise in HDL-cholesterol at day 25, in both studied groups, it seems to the levels similar to the start levels. It is only a suggestion for the presentation of the results.
Introduction and Discussion are written concisely, informative and meaningfully.
Author Response
I have read with great interest the manuscript in which the authors described the results of the SARS-CoV-2 infection influence on outcomes of metabolic, inflammatory, and liver parameters of diet-induced obese golden Syrian hamsters regarding stages of SARS-CoV-2 infection and compared to the control group of golden hamsters. The study found that obese hamsters had worse recovery than lean counterparts, and had sustained liver steatosis, inflammation, hepatocyte ballooning, and noticeable liver fibrosis. The authors concluded that in hamsters, diet-induced obesity and NASH impaired SARS-CoV-2 disease recovery. The animal model of golden hamsters was used since the SARS-CoV-2 spike protein binds to hamster ACE2, and the resulting lung disease shares several features with COVID-19 which is comparable to humans. However, in the golden Syrian hamster model, SARS-CoV-2 infection does not lead to severe clinical symptoms. Still, the authors believe that this model might be of value in characterizing the pathophysiologic mechanisms of COVID-19 and in evaluating the efficacy of treatments for the severe forms of COVID-19 observed in obese patients with NASH. Overall, the manuscript is written good, with good data and future direction.However, I have the following suggestions for manuscript improvement:
Thank you very much for this comment.
I suggest writing the manuscript in the passive form, and to exclude the “we” or “our” form in sentences. I believe that the paper could sound more scientifically if it is all written in the passive.
We have edited the ms accordingly (particularly «our» model, «our» data etc).
1.Methods-The Methods section seems vaguely presented. I suggest that the section be broken down into "Participants" (Animals) and Methods with details of the diet used and methods used for the analysis of outcome parameters. In that section, it is certainly important to include the number of tested hamsters, its total number, the number of hamsters in the intervention group and the number of hamsters in the control group. In the manuscript, there is no listed number of tested hamsters. I suppose that all hamsters had access to (and free to use) a control diet and to intervention diet (used for inducing obesity). It is not clearly stated how many hamsters were obese after 20 weeks (diet-induced obese hamsters) and how many hamsters were used for the control group. It can be suggested to display a diagram of the research flow so that the research flow can be seen.
We have now clearly indicated the number of hamsters (lean and obese NASH) in the M&M section (page 4).
2-When describing a control diet (chow) used for feeding hamsters used by SAFE Diets, Augy, France, it is stated that it consisted of 1% fat, 19.3% protein, and 55.5% carbohydrates. What were other ingredients up to 100%? I suggest stating other ingredients as "other non-energy ingredients" or something similar or recalculating the proportions of macronutrients with regard to energy content (100%). Described intervention diet has listed proportions of ingredients up to 100%, I suggest describing the control diet similarly.
Thank you for your comment. The other components are humidity 11.5%, minerals 4.6% and fiber 4%. We have amended the text accordingly in the M&M section 2.1 (page 3).
3-The phrase “dpi” (days post-infection) in Methods should be written in full meaning since there is the first time mentioned.
This is corrected (page 4)
4-In Statistical Analysis, I suggest mentioning that data were previously tested for normality using the appropriate test since the authors mention that some data were parametric and some non-parametric.
We have now indicated in the M&M section that data were previously tested for normality using the D’Agostino–Pearson-Normality-test (page 6).
5-Results-The authors presented very well their study results mostly in the form of Figures, which I very much respect. If study results were presented more in tables, with means (SD), medians (IQR) and p-values, future research can value that data presented this way. I appreciate the intention of the authors to present their data in the form of Figures, they give a picture of the rise or fall of certain values, regardless of their significant difference, but it is difficult to read the values. For example, the authors mentioned that HDL-cholesterol levels dropped on days 4 and 7 post-infection, it is difficult to see was the drop higher on day 4 or on day 7. Also, there is a rise in HDL-cholesterol at day 25, in both studied groups, it seems to the levels similar to the start levels. It is only a suggestion for the presentation of the results.
This information will be available to the reader* (now precised in the revised version). HDL-cholesterol levels (as well as LDL-cholesterol) indeed returned to basal level in both animal groups at dpi 25. This is now indicated page 11. Thank you for this remark.
*https://doi.org/10.5281/zenodo.7071688
Introduction and Discussion are written concisely, informative and meaningfully.
Thanks